# Mitochondrial cytochrome b sequence data are not an improvement for species identification in scleractinian corals

John P. Wares

Department of Genetics, University of Georgia, Athens, GA, USA

## ABSTRACT

There are well-known difficulties in using the cytochrome oxidase I (COI) mitochondrial gene region for population genetics and DNA barcoding in corals. A recent study of species divergence in the endemic Caribbean genus *Agaricia* reinforced such knowledge. However, the growing availability of whole mitochondrial genomes may help indicate more promising gene regions for species delineation. I assembled the whole mitochondrial genome for *Agaricia fragilis* from Illumina single-end 250 bp reads and compared this sequence to that of the congener *A. humilis*. Although these data suggest that the cytochrome b (CYB) gene region is more promising, comparison of available CYB sequence data from scleractinian and other reef-building corals indicates that multilocus approaches are still probably necessary for phylogenetic and population genetic analysis of recently-diverged coral taxa.

## INTRODUCTION

Coral reefs are widely recognized as being important representatives and biogenic harbors of biodiversity (*Plaisance et al., 2011*). At the same time as coral reefs are in crisis due to disease and habitat change, there is still new diversity being explored (*Breedy, Williams & Guzman, 2013*; *Breedy & Guzman, 2014*). With the incredible phenotypic diversity that may be found in an individual taxon (*Bruno & Edmunds, 1997*; *Veron, 2011*), as well as the potential for hybridization among taxa (*Vollmer & Palumbi, 2002*), biologists often turn to molecular techniques for separating taxa.

In many animal species, this approach has been relatively straightforward and has often relied on a single gene that is both highly variable at silent nucleotide positions as well as highly conserved for amino acid sequence (*Folmer et al., 1994*; *Hebert et al., 2003*). This combination allows the sequencing of this gene with universal primers, yet the discovery of tremendous amounts of nucleotide variation that may be used to distinguish taxa. However, this gene region has proven nearly useless in corals (*Shearer & Coffroth, 2008*). Researchers have also tried using other protein-coding loci such as *atp*6 (*Bongaerts et al., 2013*) to explore phylogenetic diversity in Agariciid corals, but still tended to recover nonmonophyletic taxa. Similarly, *Meyers (2013)* showed that using intron regions within the mitochondrial ND5 locus (*Concepcion, Medina & Toonen, 2006*) could not resolve many species in the genus *Agaricia*.

Corresponding author
John P. Wares, jpwares@uga.edu

The focus for such work has often been mitochondrial regions because the DNA is abundant in animal tissues, often variable within and among populations, and the lower effective size of the mitochondrial genome—a haploid genome that is typically maternally inherited—tends to result in diagnostic nucleotide characters for a population in less time than for a nuclear locus (*Avise, 2000*). For both historical and empirical reasons, some groups of systematists and population geneticists have widely used other mitochondrial regions with success. Population genetics in fishes, for example, frequently explore cytochrome b or ND4, and some have used non-coding regions (e.g., ribosomal or the D-loop origin of replication) (*Muss et al., 2001*; *Taylor & Hellberg, 2003*; *Hyde & Vetter, 2007*).

The brief goal of this study is to attempt to identify another useful mitochondrial region for population genetics and systematics studies in scleractinian corals. Here I focus on the Agariciidae; no complete phylogeny yet sufficiently resolves the endemic Caribbean genus *Agaricia* (*Meyers, 2013*; *Bongaerts et al., 2013*), and overall the family is an important one for reef development but needs further exploration of its biogeographic and phylogenetic history (*Luck et al., 2013*). Thus, this study first compares whole mitochondrial sequences between two divergent taxa of *Agaricia* (*A. fragilis* and *A. humilis*), and identifies the most divergent protein-coding region (using coding regions for increased likelihood of conserved primer development). I then analyze divergence in this region (cytochrome b, CYB) across available scleractinian data to show that this region alone is unlikely to improve our ability to separate taxa using DNA sequence-based methods.

## METHODS

To identify potential regions on the easily-sequenced mitochondrial locus, a single individual of A. fragilis (AS1943, collected in the Upper Florida Keys and detailed in *Meyers, Porter & Wares (2013)*) was shotgun sequenced with a single Illumina MiSeq library preparation as in *Wares (2013)*. Resultant single-end 250nt reads were trimmed and mapped to the *A. humilis* mitochondrial genome (GenBank DQ643831) using Geneious 7.1.4 (Biomatters). The alignment process included up to 5 iterations, with maximum gapped sites per read of 10%, maximum mismatches per read of 20% and a minimum overlap of 20 nucleotides. Annotation of this assembled genome was initiated using MITOS (*Bernt et al., 2013*) and corrected via re-alignment with the *A. humilis* sequence.

Aligned coding sequences were evaluated for K2P divergence between the two genomes using PAUP*4.0b10 (*Swofford, 2000*) as in *Shearer & Coffroth (2008)*; a sliding-window measure of divergence was calculated for 500-bp regions in 25-bp increments along the whole mitochondrial genome.

Subsequently, sequence data for the CYB locus from coral studies that included 'Scleractinia' (these studies often include other taxa from the Anthozoan subclass Hexacorallia) were downloaded from GenBank using Geneious 7.1.4 and aligned in CodonCode Aligner v4.2.2. Again, K2P distances among all sequences were obtained using PAUP*, and all pairwise distances were coded as conspecific, congeneric (excluding

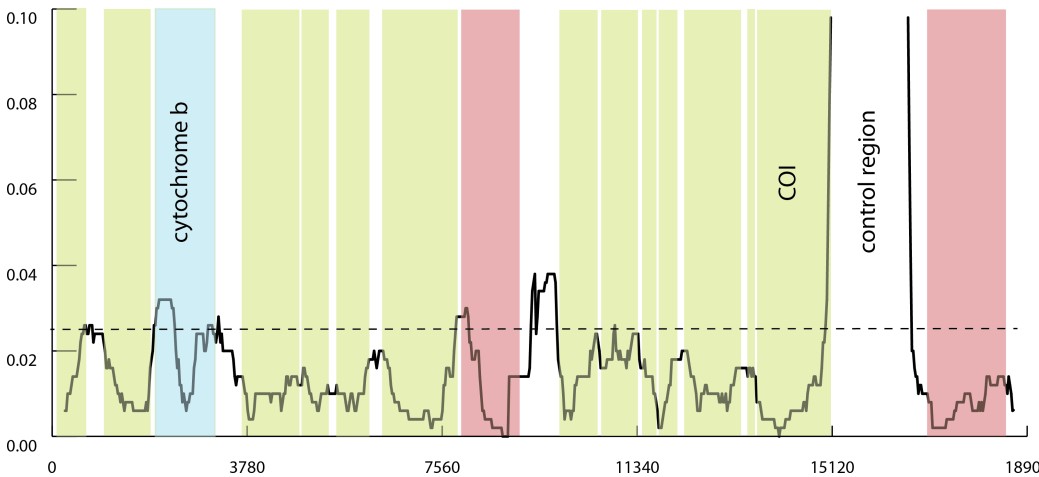

**Figure 1 Divergence of mitochondrial genomes in *Agaricia*.** Sliding window divergence between mitochondrial genomes of *A. fragilis* and *A. humilis*. Window size was 500 bp, measured every 25 bp. Coding regions are shaded in green; ribosomal regions in red. Other non-coding regions (tRNA and the intron region for ND5) not indicated. Cytochrome b is shaded blue and harbors highest mean divergence of 0.024 (dotted line).

conspecific), or "other". The distances observed for these 3 classes of comparison were density plotted using ggplot2 in the R computational environment.

## RESULTS

Illumina sequencing of the *A. fragilis* genomic DNA library resulted in a total of 31,957,468 reads. Mapping these reads to the *A. humilis* mitochondrial genome generated a single contig of 18,667 bp. The completed *A. fragilis* mitochondrial genome (Genbank KM051016) had no observed gene rearrangements and is consistent with the standard type SII for scleractinian corals (*Lin et al., 2014*).

Sliding window comparison of the two mitochondrial genomes is shown in Fig. 1. The coding region with highest divergence between the two sequences is cytochrome b (CYB) with a mean divergence of 0.024 substitutions per site. All other coding regions exhibit lower divergence per nucleotide, with COI only about 1.6 percent divergent.

Evaluation of CYB sequences across the Scleractinia and other corals for their utility in separating congeneric and more distant species included 249 pairs of intraspecific contrasts, 2,098 intrageneric contrasts, and a total of 64,981 contrasts that included 203 species from 94 genera of corals. The sequence alignment of these data is provided as File S1; a neighbor-joining tree generated from these data is provided as File S2. These contrasts, shown in Fig. 2, indicate that a divergence comparable to intraspecific diversity can be observed between members of the same genus or even more distantly related taxon pairs.

## DISCUSSION

The results of this study do less than hoped to advance molecular methods for species identification in corals. Mitochondrial DNA is often an optimal solution for metazoan species barcoding and a first attempt at species delineation (*Hebert et al., 2003*). Yet, in
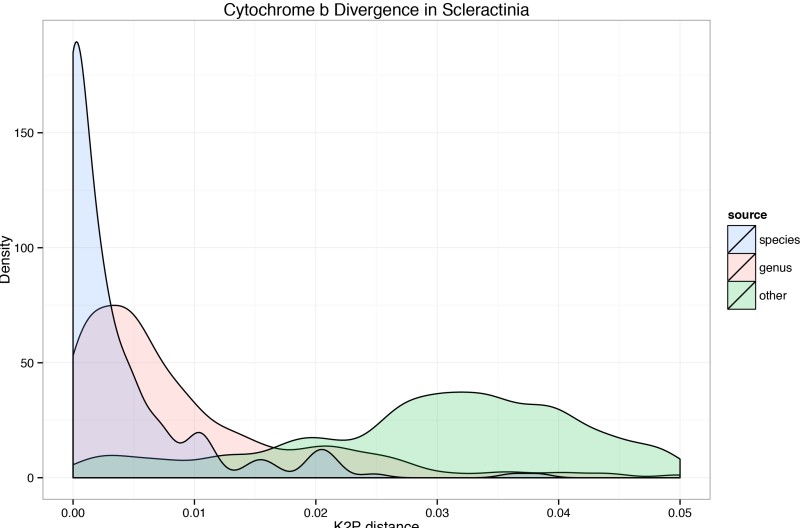

**Figure 2 Cytochrome b divergence among scleractinian and related corals.** Cytochrome b divergence, using K2P genetic distances, among taxon pairs. Plots are separated by intraspecific contrasts, intrageneric contrasts, and all other observed distances. Plot is truncated at 0.05 for clarity; all intraspecific and almost all intra-generic contrasts are shown.

corals—or anthozoans in general—the processes of mutation and DNA error correction in mitochondrial DNA (*Hellberg, 2006*), along with the propensity for hybridization among some taxa, has led to the need for more laborious locus development for such goals. Many studies are relying on microsatellite development (*Concepcion et al., 2010*; *Concepcion, Baums & Toonen, 2014*), which enables additional variation and the benefits of a multi-locus study; however, the direct identification and analysis of shared, derived characters that distinguish populations is sometimes more complicated with such data.

Ultimately the goal of species delineation is identification of character states that are diagnostic. Finding gene regions that provide sufficient information, above and beyond the variation found within a population, is the challenge. Some nuclear gene regions have shown promise. For example, *Concepcion et al. (2008)* identified the SRP54 exon-primed intron-crossing locus as being a single-copy locus that is typically more variable than non-coding regions such as the ribosomal internal transcribed spacer (ITS) regions. Other authors are combining data from several loci to attain the same goal (*McFadden, Reynolds & Janes, 2014*). Certainly it is now common to approach phylogenetic and population genetic questions with multi-locus data where possible, and the same rigor appears to be necessary for species delimitation in coral taxa.

A somewhat circular problem of using available data for consideration of barcode locus efficiency is that Genbank itself is rife with poorly identified data (*Meier et al., 2006*; *Kwong, Srivathsan & Meier, 2012*). When a nucleotide sequence is labeled/described only to genus, as in '*Discosoma* sp.', it may or may not be conspecific with *Discosoma nummiforme*. In these analyses, such taxa would be compared at the genus, not the species level; however, in the data included in this study this would affect at most 5 taxon pairs. Additionally, there is the occasional problem of species that phenotypically appear to be similar to the focal

taxon, but genetically divergent (for example, in a phylogeography study), and are labeled with '*cf.*' as indication of this uncertainty. These may or may not be the same actual species; in this study, they are treated the same as with underdescribed taxa noted above. Other taxonomic concerns among the data could similarly affect the inclusion of a contrast as within- or among-genera, but are beyond the scope of this study to resolve.

The premise of this study, that more divergent regions could be found by comparing mitochondrial genomes, is directly relevant only to the genus *Agaricia* from which these sequences derive. Using only a single genome from each species presents an incomplete picture of overall net nucleotide divergence (*Nei & Li, 1979*). However, given the typical problem of developing such markers in corals it may make sense as a general strategy to first explore available genomic data—whether mitochondrial or whole-genome—rather than blindly tackle the problem with available primer regions or use the same gene region that has proven useful in other Metazoans. Here, for example, we see that while the divergence between members of *Agaricia* is low at most protein-coding regions of the mitochondrion, the 'control region' exhibits high divergence; *Luck et al. (2013)* used another novel non-coding region to separate the Agariciid taxa *Leptoseris* and *Pavona*. It remains to be seen whether using next-generation approaches, as in this study, to generate whole mitochondrial genome sequences, may be more informative (but see *Fukami & Knowlton (2005)*) and nearly as cost-effective as attempting to capture several distinct gene regions via PCR.

## ACKNOWLEDGEMENTS

Sequence data were generated at the Georgia Genomics Facility (dna.uga.edu). The specimen of *Agaricia fragilis* was collected by MK Meyers. Thanks to WriteLatex.com for aiding my escape from the hegemony of Microsoft.

### Funding

This work was funded by NSF-EID-1015342 to John P. Wares and colleagues at the University of Georgia. The funders had no role in study design, data collection and analysis, decision to publish, or preparation of the manuscript.

### Grant Disclosures

The following grant information was disclosed by the author:
National Science Foundation, Ecology of Infectious Diseases: #1015342.

### Competing Interests

The author declares there are no competing interests.

### Author Contributions

- John P. Wares conceived and designed the experiments, performed the experiments, analyzed the data, contributed reagents/materials/analysis tools, wrote the paper, prepared figures and/or tables, reviewed drafts of the paper.

## DNA Deposition

The following information was supplied regarding the deposition of DNA sequences:
   Sequence data were generated at the Georgia Genomics Facility and deposited with GenBank, accession KM051016.

## Supplemental Information

Supplemental information for this article can be found online at http://dx.doi.org/10.7717/peerj.564#supplemental-information.

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
