# Peer review of "Mitochondrial cytochrome b sequence data are not an improvement for species identification in scleractinian corals"

_PeerJ, doi:10.7717/peerj.564_

## Round 0.1 · original submission · Major Revisions

Four reviews provide constructive criticisms that I believe you can address to improve your manuscript. In particular, expand the analysis to evaluate the multi-locus approach that is put forth in the abstract, as well as expanding on the phylogenetic analysis. There are also additional minor comments that you should take into consideration as they will help you enhance the quality and usefulness of your study.

·

Basic reporting

This manuscript describes the sequencing of the Agaricia fragilis mt genome, comparison with the A. humilis genome, and analysis of cytochrome B divergences among scleractinians (and maybe more taxa, see below). I think this is a very worthwhile study, even with the mt genome itself, noting that data for this species appear scarce. The comparison with A. humilis is also useful. So I disagree that 'the results of this study do little to advance molecular methods for species identification in corals' (Discussion, para. 1). There is much to be learnt from this paper, e.g. 'multilocus approaches are still probably necessary for phylogenetic and population genetic analysis of recently-diverged coral taxa' (Abstract; why only in the Abstract?).

I found some issues with the dataset pertaining to taxonomy and use of unidentified sequences, as well as the analysis relating to partitioning of distances and K2P correction (see below). However, if these can be addressed, the manuscript, which is well written, would be a very valuable contribution.

Experimental design

If Supplementary S1 is used to analyse and present Figure 2, then it needs a lot of cleaning up. First, the entire paper talks about Scleractinia (scleractinians), but there are other taxa in the dataset, e.g. Rhodactis, Cirripathes, Anemonia, etc. Some of these are corallimorphs, which I understand might have some evidence for being nested in Scleractinia, but if these are to be included, then the Methods need to state that explicitly.

Second, there are many unidentified taxa, like 'Faviidae sp.', 'Zoanthus_sp.' (Zoantharia!) and 'Anacropora sp.'. I'm assuming most of these go into the 'other' component (which really don't add much information since they can be variation at any taxonomic level). But for 'Anacropora sp.' sequences, for instance, are they put in intrageneric contrasts with Anacropora specimens identified to species? This can be problematic because it is possible 'Anacropora sp.' is the same species as one particular species-level Anacropora in the dataset (and thus should be intraspecific rather than interspecific). There are also 'cf.' entries in the dataset (e.g. Montipora cf. incrassata), which shouldn't be used. For how these data should be treated, see Meier et al. (2006, Syst. Biol.) and Kwong et al. (2012, Cladistics).

Third, one basic assumption in these comparisons, particularly the intrageneric ones, is that the species are assigned to the 'correct' genus. By that I mean that the genus is monophyletic, otherwise the distances will tend to be overestimated. For instance, 'Favia' favus and Favia fragum are very divergent species (Fukami et al. 2008, PLoS ONE; Budd et al. 2012, Huang et al. 2014, Zool. J. Linn. Soc). I'm assuming their divergence is grouped as an intrageneric comparison, which would be misleading. Another would be 'Favia' leptophylla and stelligera, and there are several others. Feel free to contact me for a species list with an updated generic classification based on recent phylogenetic results. WoRMS also already has most of them incorporated (http://www.marinespecies.org/aphia.php?p=taxdetails&id=1363).

Finally, the 'use of mean instead of smallest interspecific distances exaggerates the size of the "barcoding gap" and leads to misidentification' (Meier et al. 2008, Syst. Biol.), and the 'inappropriate use of Kimura-2-parameter (K2P) divergences' (Srivathsan & Meier 2012, Cladistics) is problematic. I understand that Figure 2 includes all contrasts, but if one is to make any conclusions about DNA barcoding and species delimitation, the smallest interspecific distance, uncorrected, would be more appropriate.

Validity of the findings

The data need to be cleaned up and reanalysed (see above).

Additional comments

Introduction, para. 2: 'but still tended to recover polyphyletic taxa'. I think paraphyletic is more appropriate.

Introduction, para. 4: 'PCR-based methods' is unnecessarily vague.

Methods, para. 1: Please supply collection data.

Methods, para. 3: I think it's clearer to use 'intraspecific' vs. 'congeneric interspecific' (also for Figure 2). Congeneric is vague—I hope it doesn't include intraspecific as well.

Figure 1: Please label all the coding regions for clarity.

Discussion, para. 1: 'the processes of mutation and DNA error correction in mitochondrial DNA' and 'the propensity for hybridization among some taxa' need citations. Also, 'the direct analysis of synapomorphy between populations is more complicated' needs citation and/or explanation.

Discussion, para. 3: (1) How about the 'control region'? It is hypervariable, and has been successful for resolving some groups, including agariciids based solely on that locus (Luck et al. 2013, PeerJ). (2) Please explain 'attempt to shoe-horn the coral barcoding problem in with the rest of the Metazoa'. (3) There are groups, especially cryptic species, with valid problems for barcoding, e.g. Orbicella annularis complex, for which even the entire mt genome has limited power to separate species (Fukami et al. 2005, Coral Reefs). For these, perhaps nuclear genes (PCR or whole-genome) would be more useful (see also Kitahara et al. 2014, PLoS ONE).

References: Some capitalisation issues.

·

Basic reporting

The article is clearly written and demonstrates a small but interesting study in coral phylogentics/species identification. The basic reporting criteria are met, but the manuscript is missing any sort of a data deposition or sharing plan. At the very least, the raw sequence data needs to be deposited at NCBI. A consensus sequence for the Agaricia fragilis mitogenome should also be deposited.

Minor comments:

1.) The first paragraph of the introduction needs to be expanded and clarified.
2.) This may be my personal opinion, but colloquial terms such as "unfortunately" or "hopeful" need to be replaced with more technical language.
3.) I believe that it should be interspecific and intergeneric contrasts, not intra.

Experimental design

The experimental design presented in the paper is sound and relevant to the clearly defined research question. However, I believe that the study design should be expanded substantially to better address some of the points raised in the discussion. Here are my suggestions:

1.) A phylogenetic tree of the full CYB data set or specific subsets looking at interspecies and intergeneric relationships. I do like the K2P figure, but I think that it might be good to also evaluate the CYB marker using tree based inference, as it is most likely to be used this way.

2.) Test the power of adding multiple mtDNA loci. It would be interesting to see if a concatenated marker of CYB and COI performed better, or if a combination of CYB,COI, and CR performed well. Granted, the existing sequence information might make this prohibitive, but if it is possible, it would be interesting to investigate briefly.

Minor comment:

The specific mapping parameters used in Geneious need to be specified. As it exists now, it would be difficult to directly reproduce the sequenced and assembled genome. T

Validity of the findings

No comments here.

Additional comments

Overall, I found this to be a short, simple, and interesting paper. I just came away thinking that a little more could have been done with the analysis to improve this otherwise informative manuscript. I'm recommending a major revision here, but only because my suggestions of a phylogeny and multi-marker comparison would add substantially to the results section. Otherwise, I think that these changes should be simple and quick to accommodate.

Minor comments:

Abstract-The first two sentences could be combined for clarity
"thanks to being a haploid genome that is typically maternally inherited" is a bit colloquial for my tastes
Sentence 3 of the discussion, comma after corals.

Reviewer 3 ·

Basic reporting

In this manuscript, the author determined a whole mitochondrial genome sequence of Agaricia fragilis and compared it with a different species, A. humilis, maybe to identify useful regions for species identification. Also the author compared cytb divergence among corals and got the result that cytb sequence data are not an improvement for species identification in corals.

Actually I can not understand why the author does not use more species in Agaricia. The author should investigate whether cytb is useful or not to separe species in Agaricia.
There are many data of cytb in scleractinian corals, but it is not easy to just compare them because of taxonomic and polyphyletic problems. Thus, in Agaricia, cytb data may be useful. No one knows.

In total, the author should focus on Agaricia much more.
How about control region or non-coding regions?
The author may design new primers for Agarica and related genera for several regions.

Experimental design

Actually, a whole mitochondrial genome analysis and investigatation of cytochrome b divergece are completely different topic.
to investigate cytb diveregence among scleractinian corals, a whole mitochondrial data of Agaricia is not necessary.

Validity of the findings

Useless of cytb sequence to separate species in corals is not a novel thing. As several papers showed, cytb sequence data can just separete genera partly in corals, but not species.

Only a new finding of this manuscript is that cytb is more divergent in mt coding regions between two Agaricia species.

·

Basic reporting

The manuscript from Wares “Mitochondrial cytochrome b sequence data are not an improvement for species identification in Scleractinian corals” details the findings of the lack of robustness of mitochondrial genes as barcode for hard-corals. Although the knowledge of slow evolution of mt DNA in anthozoans being not new, in my point of view, I think this manuscript is of general interest and applicability. The text is well written and I have only few comments (listed below) that may improve the manuscript. As such, once this very minor suggestions/corrections (mostly cosmetic) be addressed, I would advocate this study to be published.


-Introduction: The family Agariciidae is not endemic to the Caribbean – please check;
-I think that Scleractinian do not need to have capital “S”;
-Please italicize species names;

Hope this review was of some help.

Experimental design

No comments

Validity of the findings

No comments

---

## Round 0.2 · accepted · Accept

Thank for addressing reviewers' criticisms in a thorough fashion. I believe you have dealt properly with their major concerns.